# Uncovering the Resistance Mechanisms in Extended-Drug-Resistant *Pseudomonas aeruginosa* Clinical Isolates: Insights from Gene Expression and Phenotypic Tests

**DOI:** 10.3390/microorganisms11092211

**Published:** 2023-08-31

**Authors:** Răzvan Lucian Coșeriu, Anca Delia Mare, Felicia Toma, Camelia Vintilă, Cristina Nicoleta Ciurea, Radu Ovidiu Togănel, Anca Cighir, Anastasia Simion, Adrian Man

**Affiliations:** 1Department of Microbiology, George Emil Palade University of Medicine, Pharmacy, Science and Technology Târgu Mureș, 540142 Târgu Mures, Romania; lucian-razvan.coseriu@umfst.ro (R.L.C.); felicia.toma@umfst.ro (F.T.); camelia.vintila@umfst.ro (C.V.); cristina.ciurea@umfst.ro (C.N.C.); toganel.radu-ovidiu.22@stud.umfst.ro (R.O.T.); anca.cighir@umfst.ro (A.C.); anastasia.simion@umfst.ro (A.S.); adrian.man@umfst.ro (A.M.); 2Doctoral School of Medicine and Pharmacy, George Emil Palade University of Medicine, Pharmacy, Science and Technology Târgu Mureș, 540142 Târgu Mures, Romania

**Keywords:** *Pseudomonas aeruginosa*, carbapenemase genes, *mex* efflux pumps, Modified Hodge Test (MHT), carbapenemase inactivation method (CIM)

## Abstract

(1) Background: The purpose of the study was to describe the activity of *mex* efflux pumps in Multidrug-Resistant (MDR) clinical isolates of *Pseudomonas aeruginosa* and to compare the carbapenem-resistance identification tests with PCR; (2) Methods: Sixty MDR *P. aeruginosa* were analyzed for detection of carbapenemase by disk diffusion inhibitory method, carbapenem inactivation method and Modified Hodge Test. Endpoint PCR was used to detect 7 carbapenemase genes (*bla*_KPC_, *bla*_OXA48-like_, *bla*_NDM_, *bla*_GES-2_, *bla*_SPM_, *bla*_IMP_, *bla*_VIM_) and *mcr-1* for colistin resistance. The expression of *mex*A, *mex*B, *mex*C, *mex*E and *mex*X genes corresponding to the four main efflux pumps was also evaluated; (3) Results: From the tested strains, 71.66% presented at least one carbapenemase gene, with *bla*_GES-2_ as the most occurring gene (63.3%). Compared with the PCR, the accuracy of phenotypic tests did not exceed 25% for *P. aeruginosa*. The efflux pump genes were present in all strains except one. In 85% of the isolates, an overactivity of *mex*A, *mex*B and mostly *mex*C was detected. Previous treatment with ceftriaxone increased the activity of *mex*C by more than 160 times; (4) Conclusions: In our MDR *P. aeruginosa* clinical isolates, the carbapenem resistance is not accurately detected by phenotypic tests, due to the overexpression of *mex* efflux pumps and in a lesser amount, due to carbapenemase production.

## 1. Introduction

*Pseudomonas aeruginosa* is an ubiquitary bacteria that managed to evolve into a significant human pathogen, being involved in a diversity of infections in both immunocompetent and immunocompromised patients [1]. *P. aeruginosa* represents a continuous threat due to its intrinsic resistance to many antibiotics, and for its ability to further adapt through genetic mutation or horizontal gene transfer [2].

The treatment of *P. aeruginosa* infections usually involves the prescription of one, or a combination of antibiotics, dependent of the patient condition [3]. In severe cases, carbapenems such as imipenem and meropenem, the so-called “reserve antibiotics”, are used for therapeutic purposes, not forgetting that *P. aeruginosa* is intrinsically resistant to ertapenem [4]. But, due to its high adaptability, *P. aeruginosa* has become increasingly resistant to carbapenems and even to polymyxins, hence posing significant challenges to healthcare providers [5].

One of the main mechanisms of resistance against carbapenems involves the production of enzymes that are able to hydrolyze carbapenems, enzymes called “carbapenemases” [6]. The most common type of carbapenemases in *P. aeruginosa* belong to Ambler Class B β-lactamases, also called metallo-beta-lactamase (MβL), such as VIM (Verona Integron-Encoded MβL), SPM (Sao Paulo MβL), NDM (New Delhi MβL) or IMP (IMiPenemase) [7]. Also, Class A carbapenemases can be encountered in *P. aeruginosa,* mainly KPC (*Klebsiella pneumoniae* carbapenemase) or GES (Guiana Extended Spectrum). The GES family of enzymes were initially classified in the group of extended-spectrum beta lactamases, but later were also included in the functional group 2f of carbapenemases, due to the ability of certain point mutant derivatives to inactivate imipenem [8].

Some bacteria can easily gain plasmids carrying encoded resistance genes, raising concerns about their increased spreading, possibly leading to untreatable infections [9,10]. The genes carried on a plasmid can easily be transferred between bacterial species, leading to resistance to many antibiotics, including colistin. Colistin is often an antibiotic used as a last-resort treatment, which can become inefficient mainly due to chromosomal mutations, but possibly also due to the bacterial expression of plasmid-mediated colistin-resistance genes such as *mcr-1* [11]. Though rare in *P. aeruginosa* compared with other Gram-negative bacteria, the presence of *mcr-1* could be related with an important epidemiological outcome, due to the rapid horizontal transmission of plasmids [12]. The expression of the gene induces modification of the bacterial in the bacterial outer membrane (lipid A), preventing the antibiotic to bind and exert its bactericidal effects. The *mcr-1* gene was first described in 2015 when it was discovered in China, and it is creating difficulties in treatment management [13].

The presence of one or more resistance mechanisms, such as overexpression of efflux pumps, carbapenemase production, or impaired outer-membrane permeability, might evolve MDR (multi-drug resistant) bacteria [14]. The cellular transporters can easily remove the antibiotic from the bacterial cell before it exerts its effect, as it can be seen in the overactivity of efflux pumps [15]. The efflux pumps encountered in *P. aeruginosa* can be divided into five major families based on their structure and function: the resistance-nodulation-cell division (RND) family, the major facilitator superfamily (MFS), the ATP-binding cassette (ABC) family, the small multidrug resistance (SMR) family, and the multidrug and toxic compound extrusion (MATE) family [15,16,17]. The RND family has particular importance in *P. aeruginosa* due to the capacity of providing resistance against a broad range of antibiotics, including beta-lactams, fluoroquinolones, aminoglycosides, and tetracyclines [18]. The RND MexAB–OprM efflux pump is one of the main contributors to multiple antibiotic resistance in *P. aeruginosa*. The MexA periplasmic membrane fusion proteins coordinate the transfer of substrates across the inner and outer bacterial membrane, while the MexB protein is responsible for recognizing and binding a wide range of different molecules [19,20]. The high adaptive capacity of efflux pump activity give the bacteria an overall increased resistance, by being able to transport other substances such as detergents, disinfectants, or heavy metals [21]. 

The detection of carbapenemase production or of other resistance mechanisms might sometimes raise problems in a laboratory setting, mainly due to the lack of high sensitivity/specificity tests. According to previous studies, several phenotypic methods such as BYG Carba, CARBA NP, ROSCO tests, or the modified Hodge test can be used to detect the production of carbapenemase in *P. aeruginosa* [22]; among these, the modified Hodge (MHT) test was presented to be the most common used method [23]. 

As all laboratory procedures, MHT has a series of advantages, as well as disadvantages. The main advantages of using the MHT are: the test is inexpensive, easy to perform and provides relatively fast results. The disadvantages of using the MHT consist in: limited specificity and sensibility and subjective interpretation [24,25]. 

The mCIM test, another phenotypical method of detecting the carbapenemase production, involves the incubation of the tested strain with a carbapenem antibiotic. The mCIM test is described to be a rapid and reliable way to detect carbapenemase-producing bacteria and can be used in clinical laboratories. However, it should be used in conjunction with other laboratory tests and clinical information to make a definitive diagnosis [26,27]. 

The immunochromatographic lateral flow assays such as NG-Test Carba 5 are accurate, cheap, with short time-to-result alternative diagnostic tests, that can detect up to five common carbapenemase enzymes: KPC, NDM, VIM, IMP, and OXA-48-like. However, reduced signal or false negative results are possible due to enzyme variants with mutation in the recognized epitope [28,29].

The ability of bacteria to produce carbapenemases can be detected by molecular methods like PCR (polymerase chain reaction), a cheap, highly sensitive and specific method; though, it cannot be considered anymore a gold-standard, as it may not detect all gene variants and it is hard to cover all the spectrum of carbapenemase genes. Instead, the gold-standard for proving the presence of carbapenemase genes would be the next-generation sequencing, but this method is not suitable for clinical use, and is associated with a more complex protocol and higher associated costs [30,31]. Also, RT-PCR (reverse-transcription polymerase chain reaction) can be used to assess the gene expression of efflux pumps, which were described to have an important role in the general antibiotic resistance of *P. aeruginosa* [32]. 

The study aimed to compare the multiple methods of detecting carbapenemase, used in clinical laboratories and correlate the results with genetic findings, in order to prove the value of each method in the laboratory diagnostic protocol. A better understanding of *P. aeruginosa* resistance mechanisms plays a crucial role in the collective desire of obtaining improved treatment strategies for patients, and, implicitly, a better management of the diagnostic. 

## 2. Materials and Methods

The study was performed on clinical isolates of *P. aeruginosa* isolated from patients admitted between 2020–2023 in the Mureș County Clinical Hospital (MCCH). All isolates which met the following criteria were included in the study:-Strains identified as *P. aeruginosa*, according to the diagnostic procedures;-Only the first isolate from each patient;-Resistance to carbapenems or/and polymyxins (colistin) classes of antibiotics.

The study was approved by the Ethical Board of MCCH (no:15190).

### 2.1. Identification of Study Bacterial Strains

All the samples collected in the MCCH departments and directed to the Laboratory of Infectious Diseases were analyzed according to the common internal bacteriology protocols, depending on sample type.

The bacterial identification was made based on the morpho-tinctorial characteristics, culture, and biochemical characteristics, completed with Vitek 2 Compact (Biomerieux, Craponne, France) automatic identification.

Shortly, the primary culture was performed on Sheep Blood Agar (SBA) and Cysteine lactose and electrolyte deficient Agar (CLED Agar) (Oxoid, Basingstoke, UK) and incubated at 35 °C for 16–18 h. *P. aeruginosa* was identified as small, transparent, lactose-negative colonies, with characteristic pigment and aromatic smell. The oxidase and cetrimide test, and when the need arose, the automated method, concluded the identification. 

The antibiotic susceptibility of *P. aeruginosa* isolates was assessed by the Kirby-Bauer disk diffusion method, according to the EUCAST (The European Committee on Antimicrobial Susceptibility Testing) standard in use at that time. All the strains with “Resistant” results for meropenem disk (10 μg) were further tested using Vitek 2 (Biomerieux, Craponne, France) for evaluation of the minimal inhibitory concentration (MIC). The colistin resistance was also assessed by MIC, following microdilution protocol and EUCAST reading guides.

A number of 60 isolates corresponded to the inclusion criteria. The strains were preserved in 2 mL Tryptic-Soy-Broth (TSB) medium at −70 °C for further tests.

### 2.2. Phenotypic Detection of Carbapenemase

The phenotypic detection of carbapenemase for the 60 clinical isolates of MDR *P. aeruginosa* was performed using three different tests. The first was based on disk diffusion, namely “KPC, MBL and Oxacillinase detection in *P. aeruginosa* and Acinetobacter spp.” (KMO) (ROSCO Diagnostica, Taastrup, Denmark). The other two were based on the carbapenem inactivation principle: the modified Hodge test (MHT) and the modified carbapenem inactivation method (mCIM).

The KMO test (ROSCO Diagnostica, Taastrup, Denmark), due its name, is supposedly able to detect KPC, MBL, and oxacillinase, by using tablets impregnated with different β-lactamase inhibitors: phenylboronic acid (IMPBO), EDTA (ethylenediaminetetraacetic acid), dipicolinic acid (IMDP), also considering the activity of cloxacillin on potential AmpC producers. Actually, the test is not detecting only KPC, but also other enzymes which are part of Ambler Class A carbapenemases. All the inhibition-zone diameters were reported to the diameter for 10 μg imipenem disk provided in the kit, as recommended by the manufacturer. A negative control (*P. aeruginosa* ATCC 27853) and a positive control (*Klebsiella pneumoniae* ATCC BAA 1705) were used to validate the KMO testing procedure. The results for our tested *P. aeruginosa* isolates were interpreted according to the KMO kit specifications. 

The MHT is a classical, well-known test, used for the detection of carbapenemases in Gram-negative bacilli, and also applicable for non-fermentative ones. A 1:10 dilution of 0.5 McFarland *Escherichia coli* ATCC 25922 was inoculated on Mueller Hinton Agar, and a 10 μg meropenem disk (Oxoid, Ireland) was placed in the middle of the plate. A negative control (*P. aeruginosa* ATCC 27853), positive control (*Klebsiella pneumoniae* ATCC BAA 1705) and two *P. aeruginosa* strains to be tested were radially inoculated in straight lines from the disk. The plate was incubated for 16–18 h at 35 °C [33]. A clove-leaf shaped inhibition zone would prove the carbapenemase production by the tested strain.

The mCIM test is based on the inactivation of meropenem by bacterial-secreted carbapenemases. For this, a 10 μg meropenem antibiotic disk was immersed in 1 mL of TSB medium inoculated with the *P. aeruginosa* strain in interest and incubated for 4 h at 35 °C. A negative control using a carbapenemase-negative *P. aeruginosa* (ATCC 27853) and a positive control using carbapenemase-positive *K. pneumoniae* (BAA 1705) were also performed. Next, each disk was placed on a Muller Hinton agar medium inoculated with 0.5 McFarland meropenem-susceptible *E. coli* (ATCC 25922), and incubated for 16–18 h at 35 °C. The interpretation was made by comparing the test strains with the positive control: the absence of an inhibition zone around the meropenem disk proved the inactivation of antibiotic activity by *P. aeruginosa* carbapenemase secretion.

### 2.3. Identification of Carbapenemase Genes

For confirming the obtained phenotypical results, genetic testing was performed by endpoint PCR.

#### 2.3.1. DNA Extraction

Bacterial DNA was extracted using the boiling method: 3–4 fresh *P. aeruginosa* colonies from CLED Agar were mixed in 500 μL DN-aze free water, mixed for 10 s, and incubated in a thermomixer for 10 min at 99 °C. After cooling down for 10 min, the samples were centrifuged at 12,000× *g* rpm for 10 min, the DNA from the supernatant was quantified with a spectrophotometer (BioPhotometer D30, Eppendorf AG, Hamburg, Germany) and stored at −21 °C for further use.

#### 2.3.2. PCR for Detection of Carbapenemase Genes

For all MDR *P. aeruginosa* clinical isolates, the presence of six carbapenemase genes was evaluated by end-point PCR: *bla*_OXA48-like_, *bla*_KPC_, *bla*_IMP_, *bla*_VIM_, *bla*_NDM_ and *bla*_GES-2_. Also, the presence of plasmid-encoded colistin resistance (*mcr-1* gene) was followed. The primer sequences are presented in Table 1. The *bla*_OXA48-like_, *bla*_KPC_, *bla*_NDM_ carbapenemase genes were tested by triplex PCR, as the PCR conditions and amplicon lengths easily allowed their differentiation.

The reaction was performed in a mix consisting of 12.5 μL DreamTaq Green PCR Master Mix 2×, 0.5 μL of each reverse and forward primer (Table 1), 1 μL of DNA and water up to the final volume of 25 μL. The triplex PCR reaction was performed in the following conditions: 1 cycle of initial denaturation at 94 °C for 10 min; 36 cycles consisting in denaturation (94 °C for 30 s), annealing (57.3 °C for 40 s), extension (72 °C for 50 s); a final extension at 72 °C for 5 min. For simplex PCR, the annealing temperatures were: 52 °C for *bla*_SPM_ and *bla*_IMP_, 50 °C for *bla*_GES-2_, 59 °C for *bla*_VIM_ and 58 °C for *mcr-1*, the rest of the conditions being similar with the triplex reaction.

Positive controls were used for *bla*_NDM_ (*K. pneumoniae* ATCC BAA 2470, 438 bp), *bla*_KPC_ (*K. pneumoniae* ATCC BAA 1705, 893 bp) and *bla*_OXA48-like_ (371 bp), *bla*_IMP_ (232 bp), *bla*_VIM_ (390 bp) (well-characterized bacterial isolates from the collection of University of Medicine and Pharmacy Târgu-Mureș).

The final amplification products were loaded in 2% agarose gel (1.5 g Grade Electran^®^ DNA Agarose to 70 mL 1 × TAE) containing GelRed^®^ (Biotium Inc., Fremont, CA, USA). The electrophoresis was performed in a 1 × TAE buffer, at 100 V, for 70 min. The images were captured using MiniBIS Pro (DNR Bio-Imaging Systems Ltd., Jerusalem, Israel). The electrophoresis bands were compared with GeneRuler 100 bp (Thermo Fisher Scientific, Waltham, MA, USA) molecular ladder, and with the positive controls.

### 2.4. Evaluation of Efflux Pumps

The first 22 MDR *P. aeruginosa* preserved clinical isolates were revitalized on Sheep Blood Agar, to test their gene expression of 5 efflux pump genes: *mex*A, *mex*B, *mex*C, *mex*E and *mex*X.

#### 2.4.1. Bacterial RNA Extraction and Purification

From each fresh *P. aeruginosa* culture, 3 colonies were resuspended in 400 μL nuclease-free water and thoroughly vortexed. From this, the RNA was extracted using the Quick RNA Midiprep Kit (Zymo Research, Irvine, CA, USA), following the manufacturer protocols, obtaining 35 μL of RNA. Of these, 15 μL were treated with one unit of RNase-free DNase I enzyme (Thermo Scientific, Vilnius, Lithuania), to digest the DNA traces, before reverse transcription.

#### 2.4.2. Reverse Transcription

For all the extracts, the RNA concentration was adjusted to 300 ng/μL using nuclease-free water. The reverse transcription was performed using GoScript Reverse Transcription Kit (Promega, Madison, WI, USA), in a final volume of 25 μL composed of: 15 μL RNA 300 ng/μL, 4 μL nuclease-free water, 4 μL random primers and 2 μL reverse-transcriptase. The cDNA synthesis was performed as follows: one cycle at 25 °C for 5 min, 42 °C for 60 min, followed by 70 °C for 15 min.

#### 2.4.3. Real-Time RT PCR

The gene expression of the 5 efflux pump genes was evaluated by RT-PCR, using specific primers described in the literature, and adjusted to obtain the same annealing temperature, as presented in Table 2 [34].

A strain of *P. aeruginosa* ATCC 27853 was used as a negative control. The expression of the *rpoD* housekeeping gene was also evaluated in order to calculate the ΔΔCT and the FC (fold-change) values [16,34,35]. The PCR was performed in Applied Biosystems Quantstudio 5 (ThermoFisher Scientific, Waltham, MA, USA) thermal cycler, using GoTaq^®^ qPCR Master Mix. A 20 μL reaction mix was prepared as follows: 10 μL qPCR Master Mix 2×, 0.5 μM of each forward and reverse primer, 0.2 μL CXR (Carboxy-X-Rhodamine) as passive reference, 6.8 μL water, 1 μL cDNA. The PCR protocol consisted of one cycle of initial denaturation at 95 °C for 2 min, followed by 40 cycles of 2-step amplification (95 °C for 2 min and annealing/extension at 60 °C for 1 min). A melting curve was also used in order to check the PCR specificity.

### 2.5. Statistical Analysis

All acquired data was introduced in a spreadsheet database. Statistical analysis was performed using GraphPad InStat 3. ANOVA (with Tukey post-test) test was used to compare the gene expression of efflux pumps. The Fisher test was used to compare categorical data. The accuracy of diagnostic tests was also calculated in terms of sensitivity, specificity, accuracy, and positive (PPV) and negative (NPV) predictive values.

For RT-PCR, absolute data normalization (ΔCt) was used to compare the expression of efflux pumps within individual isolates (not normalized against the negative control *P. aeruginosa* ATCC 27853, but considering the *rpoD* activity for each isolate). Also, for relative data normalization, ΔΔCt value and FC (fold change) were calculated against *P. aeruginosa* ATCC 27853, and also considering the activity of *rpoD* housekeeping gene.

## 3. Results

From epidemiological considerations, it is always important to determine the resistance mechanism in bacteria, to be able to take actions to prevent their spreading, as well as the interbacterial resistance gene transfer. The susceptibility testing results can prove the classes of antibiotics that are inefficient against bacteria, but not the mechanism(s) of resistance. For this, supplementary tests are required, many being based on the chemical inhibition of bacterial carbapenemases.

### 3.1. Modified Hodge Test

All the 60 MDR *P. aeruginosa* isolates were tested by MHT. From the total samples, 93.33% (*n* = 56) of the samples were negative and 6.67% (*n* = 4) were considered inconclusive (a slight growth was visible, which could not be attributed to a positive result, as seen in the case of positive control—Figure 1, #4). No results were considered positive for any of the MDR *P. aeruginosa* isolates.

By comparing the MHT results with the PCR results for carbapenemase genes, it was found that the MHT test has a very low sensitivity and accuracy for *P. aeruginosa* (Table 3). This is due to the fact that in 43 isolates (71.67%), carbapenemase genes were detected, as detailed in the PCR results section.

### 3.2. KPC-MBL-Oxacillinase (KMO) Test

The results obtained with the KMO diagnostic kit according to the producer’s interpretation criteria showed that the most prevalent carbapenemase type was represented by metallo-β-lactamases (65%; *n* = 39), followed by KPC (Class A) carbapenemases (8.33%; *n* = 5) and oxacillinase (1.64%; *n* = 1). (Figure 2).

The KMO test failed to detect carbapenemase production in 26.67% (*n* = 16) of the 60 MDR *P. aeruginosa* isolates (Appendix B), even if the genetic analysis showed the presence of carbapenemase genes *bla*_GES-2_ (*n* = 10), *bla*_SPM_ (*n* = 10), or *bla*_VIM_ (*n* = 4) alone or in combination; only five of the 16 isolates with negative KMO test (31.25%) were in concordance with the PCR results. The rest of the 44 isolates (73.33%) were interpreted as positive by the KMO test. The genetic analysis indeed showed positive results for *bla*_GES-2_ (*n* = 28), *bla*_SPM_ (*n* = 3), *bla*_OXA48-like_ (*n* = 1) and *bla*_NDM_ (*n* = 1) alone or in combination; nevertheless, 12 isolates (27.27%) presented negative PCR results.

Considering all these, the statistical analysis proves a low sensitivity and accuracy of the KMO test for *P. aeruginosa* (Table 4).

### 3.3. mCIM Results

Measuring the diameters obtained for samples, compared with *P. aeruginosa* ATCC 27853, resulted in an average diameter of 27.01 mm (SD = 1.34) with a minimum value of 25 mm and a maximum of 30 mm (Figure 3c). Thus, all our tested *P. aeruginosa* strains were negative following mCIM testing. *K. pneumoniae* ATCC BAA1705 (used as positive control) managed to completely degrade the meropenem from the disk, leading to complete loss of inhibition zone (diameter read as 6 mm). For negative control, the meropenem disk incubated in presence of *P. aeruginosa* ATCC 27853 or in TSB alone (blank control), producing inhibition zones of 30 mm, respectively 36 mm diameter (Figure 3a,b).

### 3.4. Genetic Analysis Results

From the total number of 60 isolates of *P. aeruginosa*, 43 (71.67%) were found positive for carbapenemase genes, single or in combination. The triplex PCR detected one (1.66%) *bla*_OXA48-like_ positive strain and one (1.66%) *bla*_NDM_-positive isolate (Appendix A).

The other detected carbapenemase genes by simplex PCR were: *bla*_GES-2_, (63.3%; *n* = 38), *bla*_SPM_ (6.66%; *n* = 4) and *bla*_VIM_ (6.66%, *n* = 4), as presented in Appendix A. 

No isolates were found positive for the presence of *bla*_KPC_ or *bla*_IMP_.

Of the 60 clinical MDR *P. aeruginosa* isolates, three (5%) presented colistin resistance according to EUCAST, with MIC>4 mg/L. The genetic analysis for the plasmid-mediated colistin resistance *mcr-1* gene did not show any positive results.

The gene expression of *mex* efflux pumps was tested by real-time RT-PCR. The first 23 strains of *P. aeruginosa* were selected, including one PDR strain (colistin-resistant). All *Mex* genes were found with different expression levels in all tested isolates, except one, where amplification was present only for *mex*B, *mex*E and *mex*X, and not for *mex*A and *mex*C.

Absolute data normalization (ΔCt) was used to compare the expression of efflux pumps within individual isolates. Statistically significant differences (*p* < 0.0001) were found when comparing ΔCt averages of each *mex* pump (one-Way ANOVA test). The Tukey-Kramer multiple comparison post-test showed significant differences between *mex*A-*mex*B, *mex*A-*mex*E, *mex*B-*mex*C, *mex*B-*mex*X, *mex*C-*mex*E, *mex*C-*mex*X and *mex*E-*mex*X, all isolates showing a better expression of *mex*C, *mex*A and *mex*X (Table 5).

The relative data normalization compared with *P. aeruginosa* ATCC 27853 showed overexpression of *mex*B (FC > 1) in all *P. aeruginosa* isolates; *mexC* and *mex*A were overexpressed in 95.65%, respectively in 86.95%. The *mex*E and *mex*X genes showed variable expression, depending on each isolate.

The activity of efflux pumps was highly overexpressed for *mex*C in all samples but one (95.65%; *n* = 22), with a maximum FC of 162.29. On the other side, *mex*E presented the lowest activity, in 47.82% of the samples this gene being underexpressed (FC < 1) (Appendix B; Table 5).

It must be mentioned that, from those 23 patients with *P. aeruginosa-*positive isolates, 82.60% (*n* = 19) received previous treatment with antibiotics (vancomycin, amikacin, ceftriaxone, cefuroxime, penicillin, colistin, ampicillin with sulbactam or trimethoprim with sulfamethoxazole) prior of harvesting the samples, as seen from the patients’ medical charts. Fisher statistical test was used to compare the expression of *mex* genes for the *P. aeruginosa* isolates from patients with and without antibiotic treatment. A significant percentage of patients (39.13%; *n* = 9) have followed treatment with ceftriaxone. A correlation was found between the overexpression of the *mex*C gene in the patients with ceftriaxone treatment (*p* = 0.0094; OR = 20.00). The expression of the other *mex* genes was not influenced by ceftriaxone treatment (*p*-value between 0.12–1.00).

## 4. Discussion

The detection of carbapenem resistance in bacteria plays a crucial role in the establishment of treatment and also presents epidemiological relevance. Unfortunately, the number of *P. aeruginosa* carbapenemase producers has risen in the last few years.

Many studies assessed the implication of carbapenemase genes in *P. aeruginosa*, but the results are still highly variable, depending on geographic area or/and on the target population. Moreover, the multidrug-resistance phenomenon has a multifactorial etiology. The detection of carbapenemase production is a real challenge, not necessarily from a methodological point of view, but rather because the interpretation of the test might be difficult, considering the variable genetic expression of all the possibly involved resistance mechanisms. For example, a recent study conducted in 12 countries, showed that only 33% of the 807 *P. aeruginosa* strains had a positive carbapenemase phenotypical test, while 86% of them were genetically confirmed. Of those, VIM and GES were present in 54%, respectively 27% of *P. aeruginosa* strains [36,37]. In another study conducted in three medical centers from Africa between 2018–2020, from 1949 strains of *P. aeruginosa*, only 3.58% presented MβL or GES genes [38]. A Turkish study on 200 strains showed the presence of only VIM and NDM [39]. Conversely, another study conducted for 100 strains, presented that 60% were found positive for the *bla*_OXA48-like_ gene, compared to our findings, where only one strain was positive [40].

Nevertheless, our study showed GES as the most encountered carbapenemase gene in *P. aeruginosa*. This was also found in other studies, conducted in Iran, Tunisia, Egypt and Japan [41,42,43,44]. A recent review on the frequency of carbapenemase genes showed the high prevalence of *bla*_VIM_ and *bla*_IMP_ in Romania [45]. Another study in Romania, but performed mostly on wastewater-derived isolates, showed the presence of *bla*_GES-2_ in about half of the tested samples with domination of *bla*_CTX-M_ or *bla*_SHV_ [46]. All these highlight the genetic diversity of *P. aeruginosa* strains and make our study to be the first to describe a high prevalence of *bla*_GES-2_ in clinical isolate of *P. aeruginosa* MDR strains in Romania. Though, we have to mention that all our tested strains that fit the inclusion criteria were isolated from a limited area, from same clinical wards (mainly from the Dermatology Department), proving a potential local dissemination. Nevertheless, previous analysis of the same samples showed little genetic similarity, so a clonal spreading was excluded [16].

For the strains that lack carbapenemase genes, the overexpression of *mex* efflux pumps plays a crucial role in the resistance of bacteria, encountering XDR resistance [47].

In all the studies presented above, phenotypic tests were used as an initial screening for the detection of carbapenemase. Our results for the KMO test showed ambiguous results, opposingly to the genetic assessment; according to the KMO test interpretation guide, 16 strains were considered not to be carbapenemase producers, but most of them presented at least one MβL (VIM or SPM), or GES following genetic analysis. Literature suggest that tests designed for detecting phenotypic carbapenemase activity may exhibit limited effectiveness when it comes to identifying GES-2 enzymes, including false positive/negative results and low specificity [48,49]. Only 4 cases resulted in no PCR amplification, in concordance with the KMO test result. Moreover, 3 isolates supposedly to be positive for Class A Ambler (KPC) according to the KMO test result, genetically did not present *bla*_KPC_; instead, in 63.3% of the strains, *bla*_GES-2_ was detected. According to other studies, *bla*_GES-2_ is a class A Ambler which were not detected by our phenotypic tests [8]. We have shown that the accuracy for both MHT and KMO did not exceed 40%, with a sensitivity of less than 7% for both, but with good specificity. In a comparison of methods study on carbapenem-resistant *P. aeruginosa*, Peter et al. also showed that only 51.1% of a total of 133 strains were correctly identified by KMO test to present KPC and MβL, in comparison with Carba NP (in the house), when 95.5% were correctly identified as carbapenemase producers [50]. 

Regarding MHT, we consider that its high specificity is only due to the numerous negative results that were obtained. Contrarily, other studies present a very good sensibility and specificity for MHT. These discrepancies could appear due to the type of tested bacteria, as the most addressed pathogens were part of the *Enterobacterales*, not *P. aeruginosa,* or due to the subjective interpretation of the results [51]. Even though many previous studies presented the MHT as a good method to identify carbapenemase production at *P. aeruginosa* [52,53,54], formal organizations do not necessarily agree with this. For example, the standard laboratories guides such as CLSI (Clinical Laboratory Standards Institute) or EUCAST (The European Committee on Antimicrobial Susceptibility Testing) do not consider this phenotypic method for carbapenemase detection reliable, due to the difficulties in interpretation of the results or the poor sensibility and specificity [55,56]. 

The mCIM test, which is supposed to prove the presence of any carbapenemase type (without differentiating them), lead to negative results for all our tested strains. As the test was performed according to the recommendations [27,57,58], we can only assume that the negative results are due to the low-level production of carbapenemase enzymes, even if the genetic background is present in most of the isolates (carbapenemase genes). Thus, the activity of *mex* efflux pump may play a more important role in bacterial antibiotic resistance. Moreover, overexpression of *mex* efflux pumps is related to multidrug resistance, even if the trigger is only one antibiotic [59]. 

Genetic findings regarding carbapenemase production are comparable with other studies, where VIM, NDM or OXA were detected. In a study conducted in Germany, on 62 strains of *P. aeruginosa*, 2 (3.22%) presented *bla*_VIM-1_, 17 (27.41%) *bla*_VIM-2_, and one for each *bla*_GES-5_ and *bla*_IMP-82_ (1.62%) [60]. In Canada, a larger study on 3864 *P. aeruginosa* isolates, concluded that *bla*_GES-2_ was the most encountered gene (in 35% of the samples) [61]. 

The strains which cannot produce carbapenemases can evolve their carbapenem-resistance by other mechanisms to survive, such as efflux pumps. The presence of *mex* genes in *Pseudomonas* varies depending on the strain and the environmental conditions [62]. Our RT-PCR amplification results showed the presence and increased activity of efflux pumps, for all *mex* genes (tested by us), except for one sample with no activity (Ct > 40) for *mex*A and *mex*C. The results are in contradiction with some previous studies [62,63], where *mex*A was present only in half of the studied isolates, or *mexC* under 50% In our case *mex*B, *mex*E and *mex*X were presented with low activity, assuming that GES carbapenemase, in combination with other 3 pumps, actively played a role in the multiple antibiotic resistance.

We found that previous treatment with ceftriaxone was significantly associated with important overexpression of *mex* efflux pumps in MDR *P. aeruginosa*. As previous studies describe that *mex*C activity involves resistance to multiple classes of antibiotics [64,65], we can conclude that strains with overexpression of *mex*C also acquired multiple antibiotic class resistance. By searching the Pubmed database using the terms [“*mex*C efflux pump” AND “*Pseudomonas aeruginosa*” AND “ceftriaxone”], no relevant results were found. This makes our study to be the first to present the role of ceftriaxone in the overexpression of *mex*C efflux pumps in *P. aeruginosa*. This not only raises questions about the severe implication of using cephalosporins in the treatment of *P. aeruginosa*, but also emphasizes the fact that by using this type of treatment, resistance to other classes of antibiotics can be induced.

We have to mention some practical limitations in relation with the detection methods for carbapenem resistance: because the phenotypic tests are often used also for *Pseudomonas* spp., the results must be viewed with great care, the chance of obtaining erroneous results being high. Genetic tests for the expression of efflux pumps or for proving other types of resistance mechanisms including carbapenemase detection, should complement the phenotypic tests; the drawback is that these tests are expensive, do not comply with IVD (in-vitro diagnostic) marking (with few exceptions, but with significantly higher costs), are not relevant in the therapeutic decision and treatment. For these reasons, molecular testes cannot be used as routine for every suspect isolate. On the other hand, demonstrating the activity of the efflux pumps has epidemiological importance and may participate to the validation of other categories of diagnostic tests. Indeed, there are tests with higher accuracy such as CARBA NP or detection of these using MALDI-TOF, but since they are not fully implemented in many countries including Romania, many laboratories use the “classic” tests, as we analyzed in the current study.

There are also some limitations regarding the study methodology: the detection of carbapenemases genes was focused only on the most prevalent carbapenemase genes, as it is shown in the international studies on *P. aeruginosa*. Also, within GES and OXA enzymes, only *bla*_OXA-48-like_, *bla*_GES-2_ were followed due the same reasons. Because of this, some of the negative PCR results may be due to the presence of other carbapenemase genes, such as: GIM, SIM, SME, NMC, or other types of *bla*_GES_ or *bla*_OXA_, but the chance to find these in *P. aeruginosa* is very low according to literature, and would not change the statistical findings significantly. Nevertheless, this could be a reason for future and more elaborate studies. 

## 5. Conclusions

The proving of the carbapenem-resistance mechanism in *P. aeruginosa* remains a challenge in laboratories when using common phenotypic tests when a false result can mislead the therapy. Practitioners should be aware of the high adaptability of *P. aeruginosa* and the importance of other resistance mechanisms, especially as this pathogen is associated with chronic patients, nosocomial infections, and previous hospitalization background. Again, we show another side of the story of antibiotic misuse, as the usage of a common drug such as ceftriaxone can trigger more severe outcomes.

## Figures and Tables

**Figure 1 microorganisms-11-02211-f001:**
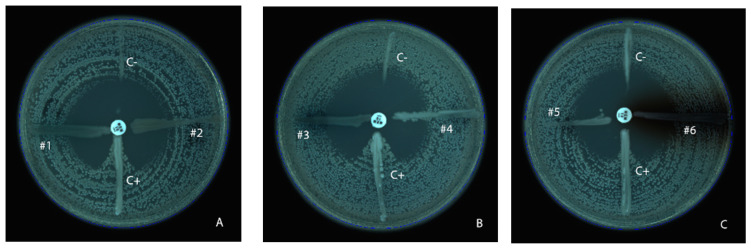
Representative results for MHT for *P. aeruginosa*. (**A**) representative negative results (#1, #2); (**B**) representative negative result (#3) and inconclusive result (#4); (**C**) representative negative results (#5, #6). (C−) negative control (non-carbapenemase-producing *P. aeruginosa* ATCC 27853); (C+) positive control (KPC-producing *K. pneumoniae* ATCC BAA 1705).

**Figure 2 microorganisms-11-02211-f002:**
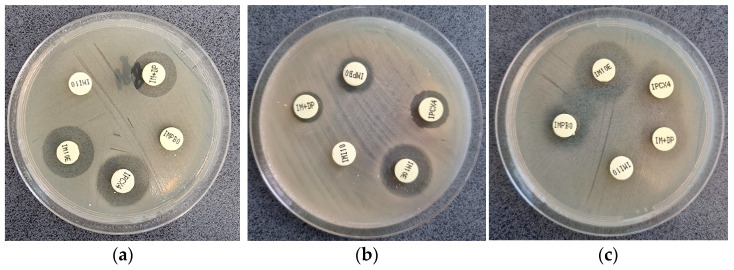
Representative images for KMO test results, according to the producer’s interpretation criteria: (**a**) no carbapenemase-producing *P. aeruginosa* (difference between IMI10 and IPCX > 5 mm); (**b**) metallo-β-lactamase-producing *P. aeruginosa* (difference between IMI10 and IM+DP > 5 mm and IMI10E > 8 mm); (**c**) KPC(ClassA)-producing *P. aeruginosa* (difference between IMI10 and IM+BO > 4 mm and IPCX4 < 3 mm).

**Figure 3 microorganisms-11-02211-f003:**
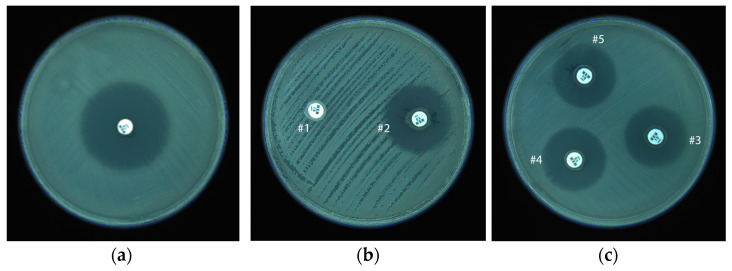
(**a**) representing blank control of a meropenem disk; (**b**) Positive control of *K. pneumoniae* ATCC BAA 1705 (#1); Negative control, the disk of meropenem incubated with *P. aeruginosa* ATCC 27853 (#2); (**c**) Negative results of tested *P. aeruginosa* isolates (#3, #4, #5).

**Table 1 microorganisms-11-02211-t001:** The presentation of the carbapenemase primers.

	Gene Type	Primer Sequence (5′ > 3′)	Amplicon Length (bp)
Triplex PCR	*bla*_KPC_ Forward	ATGTCACTGTATCGCCGTCT	893
	*bla*_KPC_ Reverse	TTTTCAGAGCCTTACTGCCC	
	*bla*_OXA48-like_ Forward	GCGTGGTTAAGGATGAACAC	438
	*bla*_OXA48-like_ Reverse	CATCAAGTTCAACCCAACCG	
	*bla*_NDM_ Forward	GGTTTGGCGATCTGGTTTTC	621
	*bla*_NDM_ Reverse	CGGAATGGCTCATCACGATC	
Simplex PCR	*bla*_GES-2_ Forward	GTTTTGCAATGTGCTCAACG	371
	*bla*_GES-2_ Reverse	TGCCATAGCAATAGGCGTAG	
	*bla*_IMP_ Forward	GGAATAGAGTGGCTTAAYTCTC	232
	*bla*_IMP_ Reverse	GGTTTAAYAAAACAACCACC	
	*bla*_VIM_ Forward	GATGGTGTTTGGTCGCATA	390
	*bla*_VIM_ Reverse	CGAATGCGCAGCACCAG	
	*bla*_SPM_ Forward	AAAATCTGGGTACGCAAACG	271
	*bla*_SPM_ Reverse	ACATTATCCGCTGGAACAGG	
	*mcr-1* Forward	CGGTCAGTCCGTTTGTTC	309
	*mcr-1* Reverse	CTTGGTCGGTCTGTAGGG	

**Table 2 microorganisms-11-02211-t002:** The presentation of the efflux pump primers.

Efflux Pump Gene	Primer Sequence (5′ > 3′)	Amplicon Length (bp)
*mex*A-Forward	ACCTACGAGGCCGACTACCAGA	252
*mex*A-Reverse	GTTGGTCACCAGGGCGCCTTC	
*mex*B-Forward	GTGTTCGGCTCGCAGTACTCGA	244
*mex*B-Reverse	AACCGTCGGGATTGACCTTGAGC	
*mex*C-Forward	ACGTCGGCGAACTGCAACG	374
*mex*C-Reverse	AGCCAGCAGGACTTCGATACCG	
*mex*E-Forward	TCATCCCACTTCTCCTGGCGC	151
*mex*E-Reverse	CGTCCCACTCGTTCAGCGG	
*mex*X-Forward	CCAGCAGGAATAGGGCGACCA	82
*mex*X-Reverse	AATCGAGGGACACCCATGCACATC	
*rpo*D-Forward	GCGGATGATGTCTTCCACCTGTTCC	132
*rpo*D-Reverse	GCGCAACAGCAATCTCGTCTGAAAGA	

**Table 3 microorganisms-11-02211-t003:** The performance of MHT for *P. aeruginosa*.

Statistic	Value	95% Confidence Indices (CI)
Sensitivity	6.00%	1.25% to 16.55%
Specificity	94.44%	72.71% to 99.86%
Positive Predictive Value	75.00%	19.41% to 99.37%
Negative Predictive Value	26.56%	16.30% to 39.09%
Accuracy	29.41%	18.98% to 41.71%

**Table 4 microorganisms-11-02211-t004:** The performance of KMO test for *P. aeruginosa,* compared with genetic detection of *bla*_GES-2_ and/or *bla*_KPC_.

Statistic	Value	95% Confidence Indices (CI)
Sensitivity	7.89%	1.66% to 21.38%
Specificity	90.91%	70.84% to 98.88%
Positive Predictive Value	60.00%	14.66% to 94.73%
Negative Predictive Value	36.36%	23.81% to 50.44%
Accuracy	38.33%	26.07% to 51.79%

**Table 5 microorganisms-11-02211-t005:** Summary of *mex* efflux pumps activity.

	*mex*A	*mex*B	*mex*C	*mex*E	*mex*X
Average ΔCt (±SD)	5.62 (±1.24)	−0.14 (±0.43)	7.71 (1.96)	1.26 (±1.17)	4.5 (±5.29)
Minimal ΔCt	3.62	−1.14	23.11	−0.42	1.04
Maximal ΔCt	7.42	0.02	11.86	3.72	15.18
Average FC (±SD)	4.17 (±2.93)	2.94 (±0.75)	37.57 (±49.65)	1.8 (±1.33)	2.27 (±1.52)
Minimal FC	0.63	2.08	0.92	0.33	0.0003
Maximal FC	11.63	4.70	162.29	4.312	4.52

## Data Availability

The data presented in this study are available in Appendix B and Appendix A.

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
