# Peer review of "Uncovering the Resistance Mechanisms in Extended-Drug-Resistant Pseudomonas aeruginosa Clinical Isolates: Insights from Gene Expression and Phenotypic Tests"

_microorganisms, 2023, doi:10.3390/microorganisms11092211_

Round 1

Reviewer 1 Report (Previous Reviewer 2)

Thank you for making the suggested recommendations.

You have answered to most of my questions, including the clarifications about the GES-2 specific primers.

It is documented in the literature that phenotypic carbaepenemase detection tests have poor performance in identifying GES-2 enzymes.  

I have certain doubts about some of the PCR products. Some amplicon lines look really faint (I have difficulties to verify some of them). A second type of test (like qPCR) would enforce the results.

There are a number of remnant English problems, like „stocked” which have not been completely changed. This can be done by the editorial theme.

Author Response

Thank you very much for the kind feedback. We are addressing your ccomments point by point.

Q: It is documented in the literature that phenotypic carbapenemase detection tests have poor performance in identifying GES-2 enzymes.

R: Thank you for this mention, we consider your remark highly relevant, and we decided to insert a phrase regarding this, in the discussion section

Q: I have certain doubts about some of the PCR products. Some amplicon lines look really faint (I have difficulties to verify some of them). A second type of test (like qPCR) would enforce the results.

R: Thank you for this observation. We agree that some bands are faint. Nevertheless, we considered the bands to be present due to the following reasons: the protocol followed exactly the previous research, their molecular weight matched the expectation, and were well defined. After zooming during the live imaging, we identified the positive bands, and marked them accordingly. Indeed, the picture may be a little under exposed, due to auto adjustment considering the molecular scale, but we don’t want to edit it using photo software.

Q: There are a number of remnant English problems, like „stocked” which have not been completely changed.

R: We change the term “stocked” with preserved and we double checked all the English.

Reviewer 2 Report (Previous Reviewer 3)

The revised manuscript seems to have met all the criticisms previously pointed out.

Author Response

Thank you very much for the kind feedback.

This manuscript is a resubmission of an earlier submission. The following is a list of the peer review reports and author responses from that submission.

Round 1

Reviewer 1 Report

This is a very interesting study that compares different phenotypic and genotypic assays. The study designs, execution, and interpretation of the results are great. The manuscript is well written. However, there are some corrections required, please find them below.

Lines 58-60 – this paragraph can be merged with the next paragraph, since both discuss about efflux pump.

Lines 124-28 – these statements are not clear. I would suggest to break them to small statements for readers to easily grasp them.

Lines 171-73  - “The reaction was performed in a final volume of 25μL, consisting of 12.5 μL DreamTaq Green PCR Master Mix 2X, 0.5 μL of each reverse and forward primers (Table 1), 1 μL of DNA and water up to the final volume.” The final volume does not add up to 25 ul, why?

Fig 2. I guess there are pos and neg discs on each plate; if they, that has to be stated in the legend.

Line 302 – “The gene expression of mex efflux pumps was tested by real-time RT-PCR. For this,..”

Lines 307-14 – “Absolute data normalization (ΔCt) was used to compare the expression of efflux pumps within each individual isolate (not normalized against the negative control P. aeruginosaATCC 27853, but considering the rpoD activity for each isolate). …” How was the normalization done?

Lines 346-7 – “Of those, VIM and GES were present in 54%, respectively and 27% of P. aeruginosa strains , respectively [27,28]. IIn 347 another study conducted in three medical centers from Africa, on a number of 1949 348 strains of P. aeruginosa, between 2018-2020, only 3.58% presented MβL or GES genes [29]” The phrase “a number of…” wrongly used in multiple places; that is not necessary. Please, delete all of them.

Lines 356-8 – “. But there are recent previous studies 356 conducted in Romania which prove the presence of other carbapenemase such as blaCTX-M, 357 blaSHVor even blaGES [37]. All these highlight the genetic diversity of” use either recent or previous….

Question. Based on your findings, which assays or assay combinations do you recommend for routine screening of clinical isolates?

I would suggest thorough review for English language.

Reviewer 2 Report

Page 2, row 82 - The test is named "Carbapenem Inactivation Method", not "Carbapenemase". - this issue is also present in the Supplimentary material.

Page 2, row 89 - one could argue that whole genome NGS would be the current gold standard, as PCR does not detect all Carbapenemase genes which may be present in a strain, nor does it offer information about gene variants.

Page 2, row 127 - "two .. phenotypic methods .. based on inactitvation of carbapenemase" - One argues that the MHT and CIM are not based on Carbapenemase Inactivation

Page 3, row - the authors proceed to name the test as CIM (carbapenm inactivation method), but proceed to describe the mCIM (use of TSB instead of sterile water, use of 4 h incubation instead of 2h).

One would also argue that the version of mCIM described by Simner et al. and in the CLSI does not use a 1mL tube of TSB, but 2 mL TSB for incubation. The implication is not clear, as the concentration of bacteria and carbapenem would be the same, but it is a deviation from the protocol that the authors do not highlight and it is unclear if this is intended.

What the authors refer to as "blaOXA" is in fact "blaOXA-48", a distinction necessary for the heterogeneous OXA family.

The authors make some unfortunate assumptions about the ROSCO test. Although it is described as the KMO diagnostic kit, the test does, not in fact, detect "KPC", but Ambler A type beta-lactamases. This is an important distinction, and creates confusion. For example, in Figure 2, the authors present a "KPC producing Pseudomonas" (presumably from their collection of tested samples), and later on state that "No isolates were found positive for the presence of blaKPC or blaIMP. "

The presence of blaGES does not imply the presence of a carbapenemase, as GES enzymes are not all carbapenemases. In absence of (targeted) sequencing data, it would stand to suppose, that some of the GES enzymes may actually be ESBLs.

The supplimentary table does not provide raw data so as to be able to check author's methodology, but it should be noted that in the text they state that "all strains were considered negative as the interpretation presented in previous studies describes the positive activity of carbapenemase with total inhibition of the meropenem disk". This is not true for mCIM for which Simner et al and CLSI offer guidance (carbapenemase positive: 6-15 mm or 16-18 mm if pinpoint colonies are present)

The authors are advised to share raw data in the supplimentary table, as one considers that there is reasonable doubt about test interpretation and peer review can not assess author interpretation.

The authors are also advised to use complementary, more modern methods and interpretations.

The article needs moderate to extensive English Language revision.

Some examples:

Page 2, row 80, "hardly" is used improperly - "difficult .. interpretation"

Page 2, row 92, "detecting carbapenemase" should be replaced with "detecting carbapenemases" or "detecting carbapenemase production"

Page 3, row 113, "where the case" should be replaced with  "when the need arose" or "when complementary tests were needed", etc.

Page 3, row 121, "were stocked" should be replaced with a correct form.

Page 3, row 143, "detection of carbapenemase production" should be used.

Reviewer 3 Report

The findings of this study could have implications for the management of patients with  infections caused by MDR P. aeruginosa, despite that fact that RT-qPCR does not seem to be a routine in the diagnosis of clinical isolates to describe the activity of mex efflux pumps. The article should discuss the feasiblity or the limitations of  using RT-qPCR to very mex expression.

Minor comments: Use of italics is missing in a few cases along the text (P. aeruginosa, mex gene names, etc